# Liquiritigenin Inhibits Lipid Accumulation in 3T3-L1 Cells via mTOR-Mediated Regulation of the Autophagy Mechanism

**DOI:** 10.3390/nu14061287

**Published:** 2022-03-18

**Authors:** Hong Qin, Ziyu Song, Chunyu Zhao, Jinxin Yang, Fan Xia, Lewen Wang, Anwar Ali, Wenya Zheng

**Affiliations:** 1Department of Nutrition Science and Food Hygiene, Xiangya School of Public Health, Central South University, 110 Xiangya Road, Changsha 410078, China; qinhong@csu.edu.cn (H.Q.); song0305ziyu@126.com (Z.S.); zcy67807189@163.com (C.Z.); yangjinxin20212021@163.com (J.Y.); xx126948@163.com (F.X.); wlwcsu@163.com (L.W.); 2Department of Epidemiology and Health Statistics, Xiangya School of Public Health, Central South University, 110 Xiangya Road, Changsha 410078, China; 206908003@csu.edu.cn

**Keywords:** liquiritigenin, obesity, lipid accumulation, mTOR, autophagy

## Abstract

Liquiritigenin (LQG) is a natural flavonoid from the herb Glycyrrhiza uralensis Fisch that exhibits multiple biological activities. However, its specific role in antiobesity and its related underlying molecular mechanisms remain unknown. The primary purpose of this study is to explore the effects and regulatory mechanisms of LQG on lipid accumulation in 3T3-L1 adipocytes. The results show that LQG significantly reduced triglyceride levels and downregulated the expression of transcription factors such as CCAAT/enhancer-binding protein α (C/EBPα) and peroxisome proliferator-activated receptor γ (PPARγ) in 3T3-L1 adipocytes. Additionally, the expression of sterol-regulatory element-binding protein 1c (SREBP1c), acetyl-CoA carboxylase 1 (ACC1), and fatty acid synthase (FASN) involved in lipogenesis was reduced by treatment with LQG. The protein expression levels of light chain 3B (LC3B), autophagy-related protein 7 (ATG7) and p62 were also modulated by LQG, leading to the suppression of autophagy. Further, LQG activated the phosphorylation of the mammalian target of rapamycin (mTOR), the inhibition of which was followed by the restored expression of autophagy-related proteins. Pretreatment with an mTOR inhibitor also reverted the expression of several genes or proteins involved in lipid synthesis. These results suggest that LQG inhibited lipid accumulation via mTOR-mediated autophagy in 3T3-L1 white adipocytes, indicating the role of LQG as a potential natural bioactive component for use in dietary supplements for preventing obesity.

## 1. Introduction

Obesity causes great public health concerns because of its association with conditions such as hepatic steatosis, dyslipidemia, and cardiovascular diseases [1]. To date, six medications, such as phentermine and orlistat, have been approved for and are commonly used for the long-term treatment of obesity [2]. However, they can cause adverse effects such as headache, insomnia, constipation, and steatorrhea [3]. Several exogenous factors, such as diet and lifestyle, can affect energy expenditure and are therefore also associated with the risk of developing obesity [4]. Therefore, it is important to explore safe and effective natural dietary phytochemicals as candidates for obesity prevention and to understand their underlying mechanisms of action.

Obesity is characterized by the expansion of white adipose tissue mass and morphological changes in the adipocytes due to excess triglyceride accumulation [5,6]. In the past several decades, studies have shown that lipid accumulation in adipose tissues is influenced by a variety of factors, including a complicated network of adipocyte-specific genes, lipogenic enzymes, transcription factors, and signaling intermediates from numerous pathways [7]. In vivo and in vitro studies have shown that peroxisome proliferation-activated receptor γ (PPARγ) is positively associated with fat-cell size in mice [8] and with obesity in humans [9]. Additionally, CCAAT/enhancer-binding protein α (C/EBPα) is well characterized as a critical transcription factor for the upstream regulator of PPARγ [10]. Moreover, sterol-regulatory element-binding protein 1c (SREBP1c) is well recognized as a key transcription factor in regulating lipogenesis and preferentially activates the transcription of genes required for fatty acid and triglyceride synthesis.

Macroautophagy is a highly regulated process whereby cytosolic proteins or whole organelles are firstly engulfed inside double-membrane autophagosomes and then translocated to lysosomes for fusion and degradation [11]. This process is customarily referred to as autophagy since it is the most well-known autophagic pathway. It was demonstrated that autophagy regulates nutrition supply and cell death and that it also plays a role in cell differentiation and development [12,13,14]. Recent research showed that, due to the removal of some intracellular components that are crucially required for cellular remodeling functions, autophagy is pivotal in the development of adipocytes [12]. This suggests that suppressing autophagy in white adipose tissue contributes to the blocking of fat depots and the amelioration of obesity [15]. It was reported that the adipose tissue of obese individuals exhibits enhanced autophagic activity [16]. Thus, interventions that influence autophagy might be a novel strategy for obesity prevention.

Liquiritigenin (LQG) is a natural flavonoid substance isolated from the herb Glycyrrhiza uralensis Fisch (also called licorice root) and has been widely used as a sweetener or antioxidant in various food items such as confectioneries, pastries, and beverages [17]. Licorice candy is popular in Europe, and teas and soups made with licorice are popular in various regions of Asia. It was shown that LQG exhibits multiple activities such as anti-inflammation, anti-hyperglycemia, and anti-neurotoxicity [18,19,20]. Studies show that Glycyrrhiza uralensis has antiobesity potential through the induction of adipocyte browning [21]. One study proved the presence of LQG in the blood samples of mice after dietary licorice-root intake [22]. However, the effects on antiobesity of LQG, the main aglycone component in Glycyrrhiza uralensis, remain unclear, and the underlying mechanisms of LQG in preventing adipose depot expansion need to be further investigated. Thus, we studied the effects of LQG on reducing lipid accumulation in adipocytes and its underlying molecular mechanisms of action. The finding of this research may provide scientific support for the use of LQG as a functional compound in dietary supplements for the prevention of obesity.

## 2. Materials and Methods

### 2.1. Chemicals and Reagents

LQG (99.49%) and rapamycin (RAP, 99.94%, an mTOR inhibitor) were purchased from Med Chem Express (South Brunswick, NJ, USA). Dulbecco’s Modified Eagle’s Medium (DMEM), penicillin–streptomycin solution, Trizol, RIPA lysis buffer, protease inhibitor, 1% phenylmethylsulfonyl fluoride (PMSF), bicinchoninic acid (BCA) protein assay kits, and thiazolyl blue tetrazolium bromide (MTT) assay kits were purchased from Ding Guo Changsheng Biotechnology Co., Ltd. (Beijing, China). Newborn calf serum (CS) was purchased from Gibco of Thermo Fisher Scientific (Waltham, MA, USA). Jiancheng Bioengineering Institute (Nanjing, China) provided the triglyceride (TG) test kits, and NCM Biotech (Suzhou, China) provided the protein loading buffer. Goat anti-Rabbit IgG (H&L)-HRP was purchased from Bioworld Technology (Bloomington, MN, USA). Antibodies against the mammalian target of rapamycin (mTOR), fatty acid synthase (FASN), PPARγ, and β-actin were purchased from Abclonal (Woburn, MA, USA). Phospho-mTOR-ser2448 (p-mTOR), light chain 3B (LC3B), recombinant autophagy-related protein 7 (ATG7), and p62 were purchased from ZEN-BIOSCIENCE (Chengdu, China).

### 2.2. T3-L1 Preadipocyte Culture and Differentiation

Preadipocytes 3T3-L1 were cultured in Dulbecco’s Modified Eagle’s Medium (DMEM) containing 10% CS and 1% penicillin−streptomycin solution at 37 °C under 5% CO_2_ atmosphere. In confluent cells, differentiation was induced as previously described [23]. The cell medium with CS was replaced with fetal bovine serum (FBS) in the first differentiation induction medium (day 0), consisting of 8 μg/mL insulin, 1 μM dexamethasone, and 0.5 mM IBMX in DMEM. After 48 h, cells were switched to the second differentiation medium, consisting of 8 μg/mL insulin only for another 48 h. Then, cells were switched to a maintenance medium of DMEM and 10% FBS to form mature adipocytes until day 8. During the differentiation process, cells were treated with or without LQG. In the experiments with the mTOR inhibitor, 3T3-L1 white adipocytes were pretreated with 20 nM RAP for 2 h before LQG treatment.

### 2.3. Cell Viability Assay

Preadipocytes were seeded in a 96-well plate at a density of 5.0 × 10^3^ cells/well and incubated until mature adipocytes formed, as described above. To determine the intervention concentrations, cells were treated with variable concentrations of LQG (0.01, 0.1, 1, 10, 25 and 50 μM) for 72 h. After the incubation period for treating cells with LQG, 20 μL MTT solution was added to each well and the cells were incubated for another 4 h. The media was then removed and 150 μL of dimethyl sulfoxide (DMSO) was added for 10 min, with shaking to fully dissolve the crystallization. Absorbance was measured at 570 nm by a microplate spectrophotometer (Thermo Scientific, Waltham, MA, USA).

### 2.4. Lipid Content Assays

The accumulation of intracellular lipids was measured and quantified using a TG assay kit and normalized to total intracellular protein using a BCA kit, following the manufacturer’s protocol. Additionally, cells were stained with Oil Red O. The cells treated with LQG were fixed with 4% paraformaldehyde for 10 min, and then stained with Oil Red O working solution for 30 min. The cells were washed with distilled water after the staining solution was removed, and the stained lipid droplets were examined under an inverted microscope (Invitrogen EVOS M7000, Waltham, MA, USA).

### 2.5. Western Blotting Analysis

The adipocytes treated with LQG and/or RAP were harvested with RIPA lysis buffer containing a protease inhibitor and 1% PMSF. The protein samples (25 µg) were separated by 8% or 10% sodium dodecyl sulfate polyacrylamide gel electrophoresis (SDS−PAGE), and then the gels were run at a constant voltage of 80 V and 120 V. Then, using a voltage of 80 V, proteins were transferred to polyvinylidene fluoride (PVDF) membranes. The membranes were incubated overnight at 4 °C with the following primary antibodies: mTOR (1:1000), p-mTOR-Ser2448 (1:1000), PPARγ (1:1000), FASN (1:750), LC3B (1:1000), ATG7 (1:1000), p62 (1:1000), and β-actin (1:300,000). Afterwards, bands were incubated for 1 h with a peroxidase-conjugated secondary antibody. Electrochemiluminescence (ECL) was used to visualize immunoreactive bands, and a chemiluminescence imager (Tanon-5500, Shanghai, China) was used to detect and measure them. Bands were quantified via densitometry by Image J 1.52a.

### 2.6. Quantitative Real-Time Reverse Transcription Polymerase Chain Reaction (RT-PCR)

Total RNA from the adipocytes treated with LQG and/or RAP was isolated utilizing a Trizol reagent and 1 μg RNA was converted to cDNA using HiScriptIIQ RT SuperMix (Vazyme, Nanjing, China). Gene expression was quantified using Hieff UNICON^®^ qPCR SYBR Green Master Mix (Yeasen Biotech, Shanghai, China) and a LightCycler 480 II (Roche, Basel, Switzerland). RT-PCR reactions were run in triplicate for each sample, and relative gene expressions were calculated using the 2^−ΔΔCT^ method after values were normalized to β-actin. The oligonucleotide primers (Sangon Biotech, Shanghai, China) used for amplification are shown in Table 1.

### 2.7. Statistical Analysis

Data were expressed as the means ± standard errors of the mean (SEM) and statistical analyses were performed with SPSS 18.0 software (Chicago, IL, USA). The comparisons among different groups were calculated by one-way analysis of variance (ANOVA) and further analyzed by independent *t*-test. When *p* ≤ 0.05, results were considered statistically significant.

## 3. Results

### 3.1. Effects of LQG on Lipid Accumulation in 3T3-L1 White Adipocytes

Firstly, the cytotoxic effects of LQG on 3T3-L1 adipocytes were determined using the MTT assay at different doses. As shown in Figure 1A, LQG at all tested concentrations did not affect cell viability. Based on the results, 10, 25 and 50 μM of LQG were used to further analyze lipid accumulation in the adipocytes. The intracellular TG levels were notably decreased when 50 μM of LQG was present in 3T3-L1 cells; however, the concentration of LQG at 25 μM only induced a tendency to reduce TG levels compared with the control group (Figure 1B). Moreover, the inhibitory effects on the accumulation of lipids by LQG were investigated by Oil Red O staining. Compared with the control group, apparent decreases in the lipid droplets in 3T3-L1 cells were observed with LQG treatments of 25 and 50 μM (Figure 1C). These results indicate that LQG treatment effectively reduced the level of lipid accumulation in 3T3-L1 cells; therefore, 25 μM and 50 μM of LQG were chosen as the experimental doses in later tests.

### 3.2. Effects of LQG on Modulating Molecular Regulators of Lipid Synthesis in 3T3-L1 Adipocytes

To explore the molecular and cellular mechanisms underlying the lipid-inhibitory actions of LQG, the expressions of c/ebpα and pparγ, the two important genes related to adipocyte development and lipid accumulation in fat cells, were examined using RT-PCR. As shown in Figure 2A,B, 50 μM of LQG significantly inhibited the mRNA expression of c/ebpα and pparγ, while LQG at 25 μM did not show obvious inhibitory effects. In addition, the protein expression level of PPARγ was also found to be downregulated by 50 μM of LQG (Figure 2F). To further determine the effects of LQG on lipid synthesis, we examined the changes in mRNA expression or protein expression in lipogenic regulators in 3T3-L1 cells. LQG at 50 μM significantly reduced the mRNA expression of srebp1c, fasn, and acetyl-CoA carboxylase 1 (acc1) (Figure 2C–E), which are genes closely associated with lipogenesis. A 25 μM dose of LQG showed a tendency to decrease the mRNA expression of srebp1c and fasn; however, there were no significant differences after statistical analysis. Additionally, FASN, a multifunctional enzyme that catalyzes the de novo biosynthesis of long-chain, saturated fatty acids starting with acetyl-CoA and malonyl-CoA, was downregulated by LQG at 25 and 50 μM (Figure 2G).

### 3.3. Effects of LQG on Autophagy-Related Protein Expression in 3T3-L1 Adipocytes

Since autophagy is a significant factor involved in adipocyte development, the expression of autophagy-related proteins after LQG treatment was examined. Endogenous LC3 is converted to LC3-I and then LC3-II during autophagosome formation. As shown in Figure 3A, the ratio of LC3BII/LC3BI was reduced after treatment with LQG. Similarly, the protein expression of ATG7 and p62 (Figure 3B,C), the substrates of specific autophagy, was decreased and increased, respectively, by LQG treatment. 

mTOR is a key regulator of cell metabolism as well as autophagy. To determine how the expression of autophagy-related proteins was affected by LQG, the effect of LQG on the protein expression of mTOR was examined using Western blotting analysis. The activation of the phosphorylation of mTOR, a major negative modulator of autophagy, was notably increased in the presence of 50 μM of LQG (Figure 3D).

### 3.4. LQG Regulated Autophagy via mTOR in 3T3-L1 Adipocytes

To investigate the correlation between mTOR and the regulation of autophagy induced by LQG, we examined whether autophagy-related proteins are modulated by RAP, which is an mTOR inhibitor. In Figure 4A, our results show that 50 μM of LQG induced phosphorylation of mTOR, whereas RAP inhibited the expression levels of p-mTOR; on the other hand, the results demonstrate that pretreatment of RAP with LQG could abrogate the activated phosphorylation of mTOR by LQG. Subsequently, expression levels of autophagy-related proteins were determined. As shown in Figure 4B,C, compared with the treatment with LQG and RAP together, treatment with LQG alone decreased the protein levels of LC3BII/LC3BI and ATG7. Meanwhile, the upregulation of p62 induced by LQG alone could be restored by the pretreatment of RAP (Figure 4D). These results suggest that the signal molecule of mTOR was involved in LQG-reduced autophagy during the process of 3T3-L1 adipocyte differentiation.

### 3.5. The Role of mTOR in Lipogenic Regulatory Factors in LQG-Treated Adipocytes

To investigate whether the effects of LQG on the process of lipid accumulation were mediated by mTOR activation, the expression of genes or proteins correlated with lipogenesis were detected by presenting LQG with RAP. As shown in Figure 5A,B, the inhibitory effects of LQG on c/ebpα and pparγ mRNA expression were abrogated by the pretreatment of RAP with LQG, which was similar to the impacts induced by RAP alone. Further, the protein levels of PPARγ (Figure 5F) were similarly modulated as their corresponding mRNA levels. The protein expression of PPARγ modulated by LQG alone was significantly downregulated compared with either the control group or the LQG+RAP group. To further elucidate whether the molecular mechanisms of the repression of intracellular lipid synthesis by LQG were correlated with mTOR, we investigated the effects of LQG on the expression of lipogenic regulatory factors by using an mTOR inhibitor. As shown in Figure 5C,D, the mRNA expression of lipogenic genes srebp1c and fasn was significantly downregulated by LQG; interestingly, RAP alone or together with LQG showed inhibitory effects on srebp1c and fasn mRNA expression as well. The acc1 mRNA-expression level was also reduced by LQG, while these effects could not be observed when LQG was presented with RAP together, or when the RAP treatment was used alone (Figure 5E), which was similar to the results for c/ebpα and pparγ mRNA expression. In order to confirm whether the effects of LQG on lipogenesis were modulated by mTOR, the protein expression levels of FASN were determined with a pretreatment of RAP with LQG. As shown in Figure 5G, the FASN protein-expression level was inhibited by LQG, and the pretreatment of RAP with LQG as well as RAP alone decreased FASN protein-expression levels compared with the control group.

## 4. Discussion

Obesity is a chronic disease that arouses great public health concern globally. Increasing attention is paid to the functions of phytochemicals, especially flavonoids, as alternative strategies in obesity prevention and amelioration. In the current study, we demonstrated that LQG administration was able to inhibit lipid accumulation in 3T3-L1 white adipocytes through modulating the genes or proteins involved in lipogenesis. Our data also showed for the first time that the function of LQG on suppressing the adipocyte accumulation of lipids occurred via the activation of mTOR, which led to a reduction of autophagy. 

Obesity triggers the development of various metabolic diseases [1,24] and is defined as an over accumulation of fat; thus, adipose tissue mass in obese individuals is expanded by a large number of lipids [25,26]. In a fed state, the principal function of white adipose tissue is lipid storage [26,27]. Our results show that 50 μM of LQG significantly reduced lipid and fat accumulation in 3T3-L1 adipocytes. There are a limited number of studies reporting LQG’s antiobesity effects. One study demonstrated that Glycyrrhiza uralensis exhibited antiobesity effects and that several substances, including the LQG in licorice extract, were able to induce UCP1 expression in 3T3-L1 adipocytes, indicating a potential role of LQ in adipocyte browning [21]. LQG is an important constituent of the flavones in Glycyrrhiza uralensis, which has shown biological properties including anti-inflammation, antihyperglycemia, antineurotoxicity, and cytoprotection [18,19,20,28]. Additionally, LQG was reported as an aglycone form in vivo [28] and could be detected in the blood samples of mice administrated licorice-root extract [22]. This current study shows that LQG is a bioactive compound in Glycyrrhiza uralensisto, which is in line with previous studies. Specifically, based on the results of this study, we proved that LQG exhibited effects on reducing lipid accumulation in adipose cells. 

Our observation of reduced TG accumulation in LQG-treated 3T3-L1 adipocytes indicates that LQG might affect the regulatory mechanisms for controlling lipogenesis during the process of adipose tissue expansion. Lipogenesis includes the process of triglyceride synthesis leading to increased fat mass, while a reduction in lipogenesis protects against the development of obesity [29,30]. In our study, we demonstrated that LQG at 50 μM significantly downregulated the mRNA expression levels of c/ebpα and pparγ, and the remarkable changes in pparγ expression in mRNA levels were consistent with the changes in its levels of encoded proteins. C/EBPα and PPARγ are two important transcription factors that have been implicated in 3T3-L1 adipocyte development, including cell differentiation and hypertrophy [5,8]. PPARγ could cooperate with C/EBP to cause mutual activation of other lipogenic genes or proteins, leading to triglyceride synthesis or lipid accumulation in cells [31]. Moreover, it is well known that de novo lipid synthesis could be decreased by targeting the fatty-acid synthesis signals of SREBP1c and its downstream molecules such as ACC and FASN. Many studies demonstrated that SREBP1c can directly bind to the acc1 promoter and the regulatory factor of the fasn gene enhancer [32], which is supported by our results. Furthermore, one study by Payne et al. [33] demonstrated that the transcription factor for C/EBPα positively regulated SREBP1c expression by using cells in which C/EBPα was inhibited with shRNA and ChIP assays. Our results clearly indicate that LQG exerted the ability to inhibit t lipogenesis by decreasing the expression levels of C/EBPα, PPARγ, SREBP1c, ACC1, and FASN. Many other natural flavonoids, such as genistein and fisetin, were also shown to suppress lipid accumulation in 3T3-L1 white adipocytes by modulating those lipogenic molecules [7,34], which is inconsistent with the results of our study. It is interesting that the protein expression of FASN was also significantly modulated at a 25 μM dose of LQG, which was not observed for the mRNA expression of srebp1c or acc1. One explanation for this is that FASN may not be regulated by only SREBP1c; even though SREBP–FASN is an important axis in the regulation of lipid metabolism, studies have proved that there could be other upstream molecular signals that modulate FASN. For example, some studies have demonstrated that the downregulation of FASN in adipocytes is partially mediated by the PKA and MAPK signaling pathways [35,36]. Therefore, even at lower doses, LQG may have a significant effect on the modulation of FASN. Additionally, the protein expression may not always be the same as the mRNA expression of its gene or relative genes, since there are complex processes of protein translation and protein post-translational modification. In general, the results that LQG suppressed de novo triglyceride synthesis suggest that LQG might be a beneficial flavonoid compound for the prevention of obesity.

Recently, increasing evidence has been put forward for the role of autophagy in the process of the accumulation of lipids in fat cells [37]. A study by Singh et al. [15] found that the inhibition of autophagy limited TG accumulation in 3T3-L1 adipocytes, indicating that the lipid accumulation in cell development had been blocked. That study also demonstrated that the knockdown of autophagy-related gene atg7 or atg5 in 3T3-L1 preadipocytes notably inhibited lipid accumulation and decreased the protein levels of regulatory factors in adipocyte development [15]. Similar effects were seen when lysosome function was inhibited pharmacologically [15]. Additionally, an adipocyte-specific mouse knockout of atg7 or atg5 generated lean mice with decreased white adipose mass and enhanced insulin sensitivity [38,39]. Although the mechanism of how autophagy influences lipid accumulation in cell development is not yet clear, a study that investigated autophagy-related protein expression at different time points proved that there was an increase in autophagy on day 6 after 3T3-L1 adipocyte differentiation, as evidenced by the significantly increased expression of LC3BII/LC3BI and ATG12 [40]. LC3B is a well-known autophagic marker during the whole process of autophagy, and ATG7 is an essential protein for the induction of autophagosomes. In the present study, we found that LQG was able to inhibit autophagic activity, since it was observed that the autophagy-related protein expression levels of LC3BII/LC3BI and ATG7 were significantly repressed. In addition, p62 was significantly increased after LQG treatment. It is commonly recognized that p62 is responsible for the cargo selection and transport of protein aggregates, and it is degraded together with its cargo in lysosomes. The loss of atg genes may block the fusion of autophagosomes with lysosomes, leading to an increase in p62 protein expression [41]. Thus, the level of p62 is usually inversely correlated with autophagy. In this study, the expression of p62 was significantly increased after LQG treatment. These findings add to growing evidence that administration of LQG may inhibit autophagy.

The phosphorylation expression of mTOR, an autophagy inhibitor, was significantly activated by LQG treatment. Many published studies have demonstrated that mTOR is a key regulator of autophagy [40,42]. Zhang et al. [43] showed that LQG protects liver tissue by enhancing mTOR-mediated autophagy in arsenic-trioxide-induced liver injury. The inconsistent results for LQG modulating mTOR expression and autophagy might be due to different tissues being under-investigated such that the role of autophagy in adipocyte development would differ from that in hepatic cell damage [44]. Additionally, the mechanisms correlating with autophagy in different periods of disease development could also differ. One study showed that arsenite exposure reduced the differentiation of murine brown adipocytes via autophagy inhibition in brown adipose tissue [45]. However, several other studies demonstrated that the suppression of autophagy reduced lipid accumulation or induced brown-like adipocyte formation in adipocytes and adipose tissue [15,38,39,40], which was inconsistent with our results. The current study shows that the inhibition of mTOR by RAP significantly decreased the p-mTOR expression activated by LQG. On the one hand, this supports the assumption that mTOR was stimulated in the LQG modulation of 3T3-L1 adipocyte development, and on the other hand, it shows that RAP principally regulated mTOR when treated together with LQG. A study investigating resveratrol also proved that polyphenols reduced lipid accumulation in 3T3-L1 cells by activating mTORC1 and its downstream site p70S6 [46]. Another polyphenol, raspberry ketone, was confirmed to considerably stimulate the expression of p-mTOR and inhibit the expression of SIRT1 on day 6 of 3T3-L1 adipocyte differentiation, resulting in a reduction in lipid accumulation [40]. 

mTOR also plays a central role in regulating fat and lipid homeostasis. The inhibition of LQG in the gene or protein expression of C/EBPα, PPARγ and ACC1 was abrogated by pretreatment with RAP, demonstrating that mTOR was involved in the lipid synthesis process affected by LQG. Considering the role of mTOR in the autophagy induced by LQG, this suggests that LQG inhibited lipogenesis in 3T3-L1 white adipocytes through mTOR-mediated autophagy. Some studies have shown that the inhibition of mTORC1 signaling genetically impairs lipogenesis by regulating the SREBP transcriptional network via S6K1 or Lipin1 [47]. However, the regulating network of mTOR in lipid homeostasis is very complex and shows crosstalk with a variety of molecules, including different substrates of mTOR. For example, Lipin1, which could be inhibited by mTORC1, was shown on the one hand to negatively regulate SREBP1c [48] and was found, on the other hand, to act as a key factor in adipocyte maturation and maintenance by stimulating PPARγ [49]. Thus, mTOR activation might lead to the suppression of PPARγ, which supports the results of the mTOR and PPARγ expression patterns in our current study. Moreover, even though mTOR is often recognized as a stimulator of SREBP in research regarding lipid metabolism [50,51], it is notable that mTORC1 signaling is essential, but not sufficient, to activate SREBP1c. For example, C/EBPα was also able to regulate expression of SREBP1c [33], indicating a parallel regulation of C/EBPα, PPARγ, and SREBP1c by the autophagic pathway via mTOR, which was in line with the results of LQG in our current study. Interestingly, the current study only shows that the co-treatment of LQG and RAP diminished the effects induced by LQG, while RAP alone did not result in significant opposite effects. Some studies have shown that RAP inhibited C/EBPα or PPARγ protein expression [52], which could not be observed in our data. The inconsistent results may be due to different concentrations and distinct stages of cell development in which RAP was added to treat the cells. In contrast, RAP treatment alone significantly downregulated the protein and mRNA expressions of SREBP1c and FASN in our results, showing similar impacts to those induced by LQG. This indicates that RAP, an mTOR inhibitor, regulates the expression of SREBP1c and FASN through different signaling pathways from LQG, which may not involve the modulation of autophagy. It was reported that RAP was able to inhibit lipogenesis via the SREBP1c–FASN axis [53,54], which was also observed in our results. Although the mechanisms of LQG and RAP in modulating 3T3-L1 adipocyte lipogenesis might differ, the results from pretreatment of RAP with LQG demonstrate that LQG modulated mTOR activity. Moreover, the activation of mTOR was correlated with reductions in the autophagy and lipid-synthesis processes. This also suggests the important role of mTOR in the molecular mechanisms of LQG in modulating adipocyte lipid accumulation.

## 5. Conclusions

LQG exhibited a significant reduction in lipid accumulation in 3T3-L1 white adipocytes in our current study. The positive effect was associated with the mTOR-mediated autophagy process (Figure 6). Together, our findings make an essential contribution to understanding the biological activities of LQG. This present study provides scientific evidence for supporting the utilization of LQG as a bioactive flavonoid to prevent obesity by inhibiting the accumulation of lipids in fat cells. This in vitro study provides evidence supporting further research regarding the properties of LQG with respect to preventing adipose tissue expansion in in vivo animal studies or even in human clinical studies. LQG can be expected to be developed as a new functional compound for application in dietary supplements. Moreover, the study supports the crucial role of autophagy in 3T3-L1-cell lipid accumulation. In addition to the bioactive phytochemicals from herbs, the molecular signals correlated with autophagy regulation might be novel and efficient targets for preventing obesity in the future.

## Figures and Tables

**Figure 1 nutrients-14-01287-f001:**
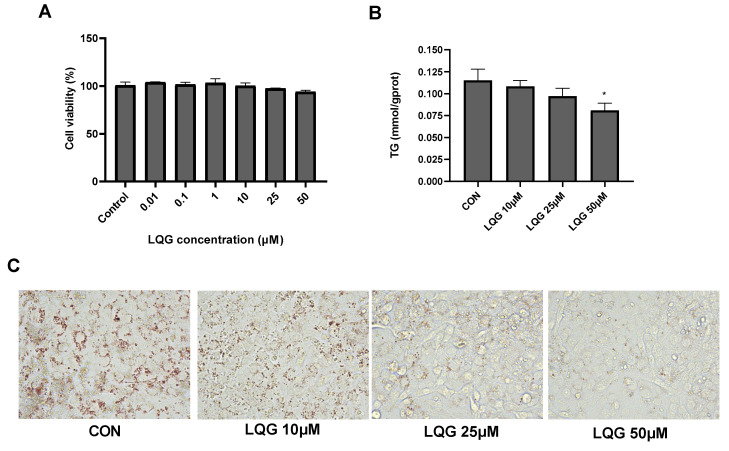
Effects of liquiritigenin (LQG) on cell viability and lipid accumulation in 3T3-L1 adipocytes. (**A**) 3T3-L1 cells were treated with different concentrations of LQG (0, 0.01, 0.1, 1, 10, 25 and 50 μM) and cytotoxicity of the cells was expressed as optical density percentage. (**B**) LQG induced suppression of intracellular triglyceride (TG) levels in 3T3-L1 cells. TG levels of LQG at 10, 25 and 50 μM were assayed and are expressed in bar charts. (**C**) Mature 3T3-L1 adipocytes treated with 10, 25 and 50 μM LQG were stained with Oil Red O to observe lipid droplets at 400× magnification. Data are presented as means ± SEMs and analyzed with one-way ANOVA (*n* = 3). * marks significant differences compared with control group (*p* ≤ 0.05).

**Figure 2 nutrients-14-01287-f002:**
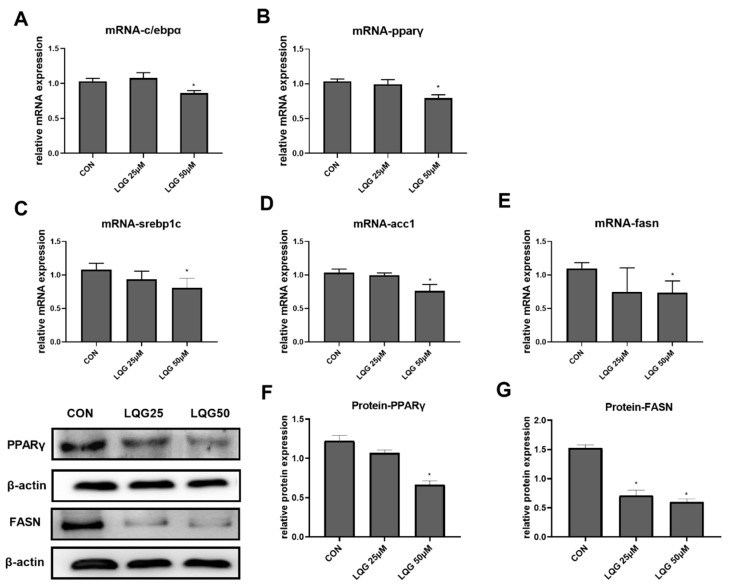
Effects of LQG on molecular regulators of lipid synthesis in 3T3-L1 adipocytes. The cells were treated with 25 and 50 μM LQG. The mRNA-expression levels of (**A**) c/ebpα, (**B**) pparγ, (**C**) srebp1c, (**D**) acc1, and (**E**) fasn were measured by RT-PCR, and the protein levels of (**F**) PPARγ and (**G**) FASN were analyzed by Western blotting. Protein expression was quantified by densitometry, and the relative intensities are expressed in the bar charts. Values are presented as means ± SEMs (*n* = 3) and analyzed with one-way ANOVA. * shows a statistically significant difference compared with control group (*p* ≤ 0.05).

**Figure 3 nutrients-14-01287-f003:**
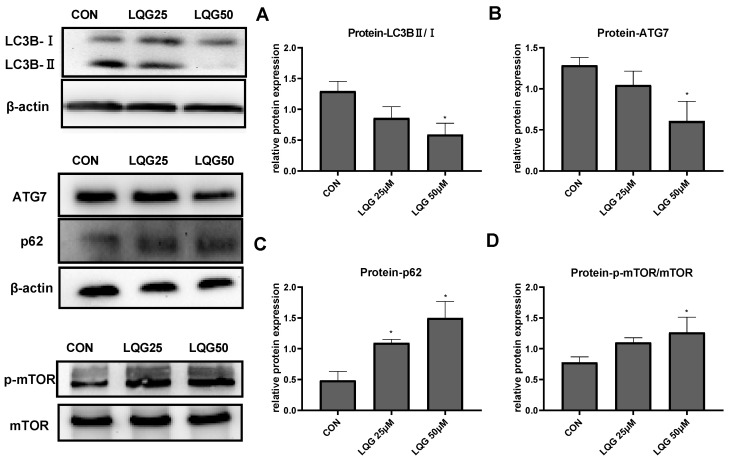
Effects of LQG on autophagy-related protein expression in 3T3-L1 adipocytes. The cells were treated with 25 and 50 μM LQG. The protein expression levels of (**A**) LC3B, (**B**) ATG7, (**C**) p62, and (**D**) phosphorylated-mTOR (p-mTOR) were analyzed by Western blotting. The results are means ± SEMs of three independent experiments and analyzed with one-way ANOVA. * marks significant differences compared with control group (*p* ≤ 0.05).

**Figure 4 nutrients-14-01287-f004:**
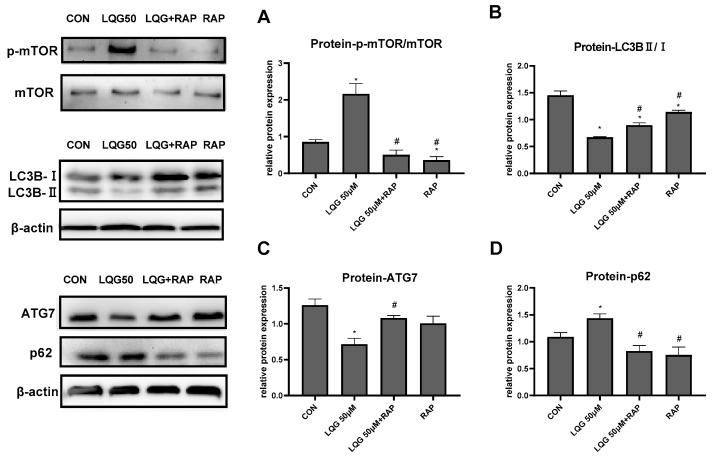
Effects of rapamycin (RAP) on modulating autophagy-related factors in LQG-treated 3T3-L1 cells. The cells were divided into four groups: control group, 50 μM LQG group, LQG + RAP group, and RAP-alone group. Protein expression of (**A**) p-mTOR and mTOR, (**B**) LC3BII/LC3BI, (**C**) ATG7, and (**D**) p62 were quantified by densitometry, and the relative intensities are expressed in the bar charts. Western blotting bands represent detection of the proteins from three independent tests. Data are presented as means ± SEMs and analyzed with one-way ANOVA. * vs. control group (*p* ≤ 0.05); # vs. 50 μM LQG-alone treatment group (*p* ≤ 0.05).

**Figure 5 nutrients-14-01287-f005:**
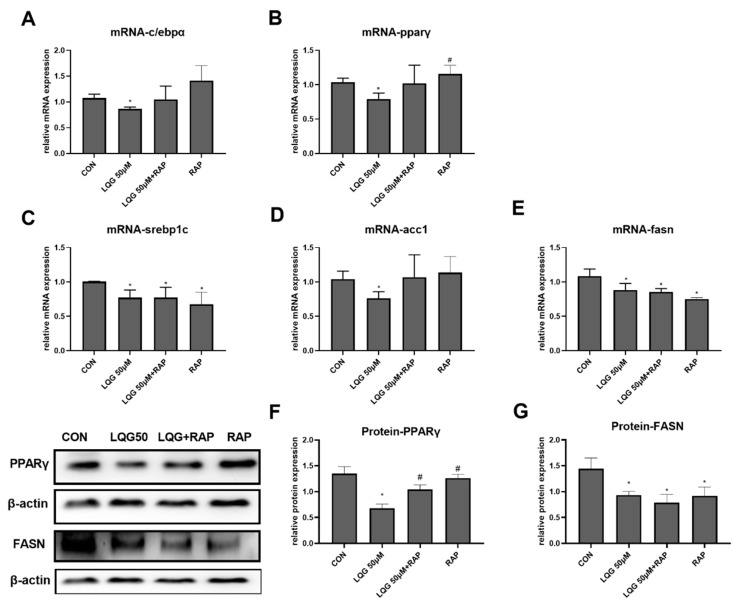
Effects of RAP on modulating lipogenic factors in LQG-treated 3T3-L1 cells. The cells were divided into four groups: control group, 50 μM LQG group, LQG + RAP group, and RAP-alone group. The mRNA expression levels of (**A**) c/ebpα, (**B**) pparγ, (**C**) srebp1c, (**D**) acc1, and (**E**) fasn were measured by RT-PCR and the protein levels of (**G**) PPARγ and (**F**) FASN were analyzed by Western blotting. Protein expression was quantified by densitometry and the relative intensities are expressed in the bar charts. Western blotting bands represent the detection of the proteins from three independent tests. Data are presented as means ± SEMs and analyzed with one-way ANOVA. * vs. control group (*p* ≤ 0.05); # vs. 50 μM LQG-alone treatment group (*p* ≤ 0.05).

**Figure 6 nutrients-14-01287-f006:**
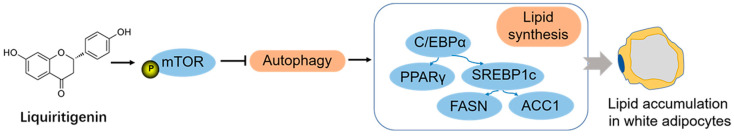
LQG reduced lipid accumulation in 3T3-L1 white adipocytes via mTOR-mediated autophagy mechanism.

**Table 1 nutrients-14-01287-t001:** Primer Sequences.

Name	Primer Sequence (5′ to 3′)
pparγ	forward: TCTCCCCACATCCTTTCT
	reverse: CTGCCGTTGTCTGTCACT
c/ebpα	forward: CTGGAAAGAAGGCCACCTC
	reverse: AAGAGAAGGAAGCGGTCCA
srebp1c	forward: GCAACACAGCAACCAGAA
	reverse: GAAAGGTGAGCCAGCATC
acc1	forward: AAAACAGGGAGGAAGCAA
	reverse: TCACCCCGAATAGACAGC
fasn	forward: GCCCAAGGGAAGCACATT
	reverse: CGAAGCCACCCAGACCAC
β-actin	forward: TTGCGTTACACCCTTTCT
	reverse: ACCTTCACCGTTCCAGTT

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
