# Peer review of "Liquiritigenin Inhibits Lipid Accumulation in 3T3-L1 Cells via mTOR-Mediated Regulation of the Autophagy Mechanism"

_nutrients, 2022, doi:10.3390/nu14061287_

Round 1
Reviewer 1 Report
It is a good one, innovative on the effect of liquiritigenin, it is well written. the test and data presentation are clear.
Reviewer 2 Report
Comments for the Authors:
This paper by Qin and colleagues has examined the effect of Liquiritigenin (LQG) on 3T3-L1 adipocytes for potential insights into the mechanisms by which it might act as an anti-obesity drug. Their results suggest that this compound may inhibit lipid accumulation and adipogenesis via 26 mTOR-mediated autophagy during differentiation of the 3T3-L1 adipocytes in culture. The major assumption underlying these studies is that adipocyte differentiation plays an important role in the development of obesity. As the authors know, the overfeeding studies of Sims et al years ago (J Clin Invest. 1971 May;50(5):1005-11) showed that acute weight gains were produced by increasing the size of existing adipocytes, NOT an increase in the number of fat cells. It thus seems that the underlying premise of this study is invalid from this reviewers perspective.
Specific Comments:
Title: OK as written
Abstract: OK as written
Introduction:
Line 39: It is debatable whether “natural products” play a crucial role in potential treatments. Please modify this sentence to make it less strong.
Line 42: Most obesity is “hypertrophic” not hyperplastic and thus adipogenesis is not usually a key factor, rather enlargement of already existing cells. Please rewrite for accuracy.
Line 72: I don’t think that a study in castrated female mice is very good evidence for the clinical use of any drug. Please eliminate this nonsense. It certainly doesn’t add to my belief that the flavonoid you are talking about has any clinical use.
Methods and Materials: Clearly written.
Results:
Figure 1 and 2: It appears that only the highest dose of 50 uM was biologically effective. It would have been interesting to have a higher dose to test for further effects.
Figure 3: The only place where a lower dose was effective in this set of figures was in the relative protein expression for FASN, but not in any of the other relative mRNA expression measurements.
Discussion:
Line 314: As noted above you don’t have to inhibit lipogenesis to deal with obesity and although the studies are well presented, it is unclear what their relevance to obesity may be, if any.
Line 320: Although you have demonstrated that LQC inhibits adipogenesis, it not clear that adipogenesis plays much of a role in the accumulation of lipid that is the hall mark of obesity. Rather simply enlargement of existing cells would be sufficient.
Line 326: As noted above a study of castrated female mice is not very helpful for insights into obesity.
Line 348: The demonstration of effects of LQC on lipid accumulation may be a way to refocus this paper on a message that would be relevant to most individuals with obesity.
Line 360: Blocking autophagy as a mechanism for preventing fat cell accumulation of lipid might also have some validity a focal point for the effect of LQC.
Conclusions:
Line 450: The authors say “This present study provides scientific evidence for supporting the utilization of LQG, as a bioactive flavonoid, to help lose weight by inhibiting the differentiation process of white adipocytes”. As I noted earlier, most obesity occurs by “hypertrophy” of existing cells, not differentiation of new ones. If the only use of LQC is to inhibit differentiation of new adipocytes its value for treatment human beings with obesity would be limited.
Figures: The figures are very clear.
Writing: Would benefit have someone who was a primary English speaker edit the text.
Reviewer 3 Report
The paper attempted to prove the anti-obesity efficacy and mechanism of action of LQG.
However, several aspects must be supplemented to achieve the research purpose.
1. To prove the anti-obesity effectiveness of LQG, LQG should be administered oral to obese animals to show weight loss and significant changes in major biomarkers.
2. As a result of analyzing the LQG content of Glycyrrhiza uralensis extract, it should be shown that LQG is a major component.
Round 2
Reviewer 2 Report
Revised version is satisfactory
Reviewer 3 Report
The author did not show any improvement in the revised manuscript through experiments.